# DUAL-MODEL DEFENSE: SAFEGUARDING DIFFUSION MODELS FROM MEMBERSHIP INFERENCE ATTACKS THROUGH DISJOINT DATA SPLITTING

## ABSTRACT

Diffusion models have demonstrated remarkable capabilities in image synthesis, but their recently proven vulnerability to Membership Inference Attacks (MIAs) poses a critical privacy concern. This paper introduces two novel and efficient approaches (DualMD and DistillMD) to protect diffusion models against MIAs while maintaining high utility. Both methods are based on training two separate diffusion models on disjoint subsets of the original dataset. DualMD then employs a private inference pipeline that utilizes both models. This strategy significantly reduces the risk of black-box MIAs by limiting the information any single model contains about individual training samples. The dual models can also generate "soft targets" to train a private student model in DistillMD, enhancing privacy guarantees against all types of MIAs. Extensive evaluations of DualMD and DistillMD against state-of-the-art MIAs across various datasets in white-box and black-box settings demonstrate their effectiveness in substantially reducing MIA success rates while preserving competitive image generation performance. Notably, our experiments reveal that DistillMD not only defends against MIAs but also mitigates model memorization, indicating that both vulnerabilities stem from overfitting and can be addressed simultaneously with our unified approach.

## 1 INTRODUCTION

In recent years, diffusion models (Sohl-Dickstein et al., 2015; Ho et al., 2020; Rombach et al., 2022) have rapidly emerged as a powerful tool for image generation, outperforming traditional methods such as Generative Adversarial Networks (GANs) (Goodfellow et al., 2014) and Variational Autoencoders (VAEs) (Kingma, 2013). These models, including well-known examples like Stable Diffusion models (Rombach et al., 2022; Podell et al., 2023), DALL-E 2 (Ramesh et al., 2022) and Imagen (Saharia et al., 2022), utilize a progressive denoising process that results in higher-quality and more stable image generation compared to previous architectures. By gradually transforming random noise into clean images, diffusion models excel at producing detailed and realistic visuals across various applications, from graphic design to medical imaging.

However, the superior performance of diffusion models relies heavily on large and diverse datasets, which often include sensitive information such as copyrighted images, personal photos, medical data, and even stylistic elements from contemporary artists. The nature of these datasets poses significant privacy risks, as diffusion models can inadvertently memorize and reproduce parts of their training data during the generation (Carlini et al., 2023). This replication of training data during inference makes diffusion models vulnerable to Membership Inference Attacks (MIAs) (Shokri et al., 2017; Matsumoto et al., 2023; Wu et al., 2022), which aim to determine whether specific samples are present in their training data. If a model has been trained on sensitive datasets, an attacker might extract or infer specific details about the data used in training, leading to unintended exposure of private or proprietary information.

Therefore, implementing robust defense mechanisms to protect against MIAs and other privacy-related attacks is crucial. Existing defense methods for MIAs, such as those based on model distillation (Tang et al., 2022; Shejwalkar & Houmansadr, 2021; Mazzone et al., 2022), have proven effective in image classification models by reducing overfitting and limiting memorization of train-

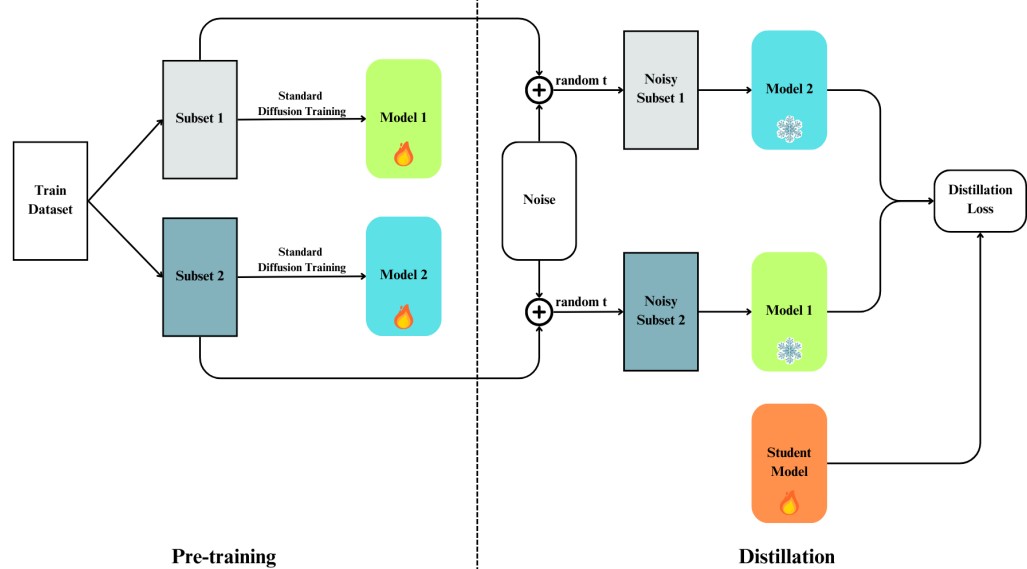

Figure 1: Our proposed defense method DistillMD with model distillation. (**Left**): We divide the training dataset into two non-overlapping subsets. Each subset is then used to train a separate diffusion model with the vanilla diffusion loss. (**Right**): During the distillation phase, for each iteration, if a data point belongs to subset 1, it is passed through the pre-trained model 2 (which is frozen) to generate a "soft" label. Similarly, if a data point belongs to subset 2, it is passed through the pre-trained model 1 (also frozen) to produce a "soft" label. The student model then uses this "soft" label as the target to compute the diffusion loss.

ing data. However, these approaches cannot be directly applied to diffusion models due to their unique structure and the resource-intensive nature of distillation processes, especially in large diffusion models.

To address these challenges, we propose a tailored distillation method optimized for diffusion models, namely DistillMD (see Fig. 1), which is computationally efficient and effective in preventing MIAs. Compared to other distillation defenses, one key advantage of our method is that it does not require additional test data for the teacher to produce non-member labels. This limitation of other approaches hinders their applications in cases where we do not have much data to train and evaluate the models. To evaluate this benefit, we perform our defense in the model fine-tuning paradigm with a small dataset in Section 4.2.

While effectively alleviating any attack, the distillation method often requires a high computational cost to train a student model, hindering the method's application to resource-constraint settings. For resource-constrained environments where the overhead of model distillation is impractical, we propose another dual-model defense (DualMD) method that does not require additional training other than the two teacher models but can still efficiently mitigate MIAs in black-box settings. The method is illustrated in Fig. 2.

Although the mentioned techniques can be effective for unconditional diffusion models, they can fail to protect conditional diffusion models due to the strong overfitting to the conditions. For example, Pang & Wang (2023) designed their attack to exploit this property using text prompts to guide diffusion models to produce images in a distribution close to the target images.

Similar to MIAs, model memorization is also related to model overfitting, and prompt overfitting has been extensively studied in diffusion model memorization. For example, Somepalli et al. (2023b) observed that prompt overfitting plays a crucial role in model memorization and proposed several techniques to reduce the effect. Wen et al. (2024) further argued that some tokens can be more important than others to guide the generation. In Section 4.2, we show that DualMD and DistillMD alone cannot effectively defend against attacks utilizing the text guidance and propose a technique

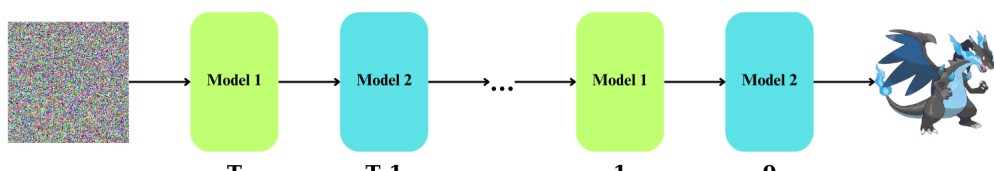

Figure 2: The efficient defense method DualMD with modified inference pipeline. The two models, which are trained on disjointed subsets, are used to denoise images alternately.

following Somepalli et al. (2023b) to diversify the training prompts. Although this straightforward approach draws inspiration from model memorization, it appears to be essential in defending against MIAs, as demonstrated in Section 4.2. Furthermore, in Section 3.6, we highlight that implementing membership inference defenses can effectively mitigate model memorization. This underscores a significant and inherent connection between these two areas.

We summarize our contributions as follows:

- We propose two mitigation strategies to defend against Membership Inference Attacks (MIAs): DualMD, targeting black-box attacks via an inference-only approach, and DistillMD, which defends against both white-box and black-box attacks through a distillation-based method. Our evaluation reveals that the distillation approach is more suitable to maintain high generation quality of unconditional diffusion models, while dual-model inference better preserves the quality of text-to-image diffusion models.

- We evaluate the effectiveness of our methods in training large text-to-image diffusion models and propose a technique to prevent the models from overfitting to the prompts. Our experiments demonstrate that we can significantly reduce the risk of personal data leakage in both white-box and black-box settings.

- We show that memorization mitigation techniques can be applied to defend MIAs and that defending against MIAs can mitigate model memorization. To the best of our knowledge, we are the first to establish this bidirectional connection between these two areas.

## 2 BACKGROUND AND RELATED WORK

**Diffusion Models**   Recent breakthroughs in diffusion models have demonstrated remarkable success across various generative tasks. As powerful generative models, diffusion models (Sohl-Dickstein et al., 2015; Ho et al., 2020) produce fascinating images by progressively denoising inputs. They first incorporate noise into data distributions through a forward process, then reverse this procedure to recover the original data. In particular, starting with an initial data original image $\mathbf{x}_0$ sampled from a (unknown) distribution $q(\mathbf{x}_0)$, the forward process gradually diffuses $\mathbf{x}_0$ into a standard Gaussian noise $\mathbf{x}_T \sim \mathcal{N}(\mathbf{0}, \mathbf{I})$ through $T$ consecutive timesteps, where $\mathbf{I}$ is the identity matrix. Specifically, at timestep $t \in \{1, \ldots, T\}$, the diffusion process $q(\mathbf{x}_t|\mathbf{x}_{t-1})$ and the denoising process $p_\theta(\mathbf{x}_{t-1}|\mathbf{x}_t)$ are defined as follows:

$$q(\mathbf{x}_t|\mathbf{x}_{t-1}) = \mathcal{N}(\mathbf{x}_t; \sqrt{1-\beta_t}\mathbf{x}_{t-1}, \beta_t\mathbf{I}),$$
$$p_\theta(\mathbf{x}_{t-1}|\mathbf{x}_t) = \mathcal{N}(\mathbf{x}_{t-1}; \boldsymbol{\mu}_\theta(\mathbf{x}_t, t), \Sigma_\theta(\mathbf{x}_t, t)), \tag{1}$$

where $\beta_t \in (0, 1]$ is an increasing noise scheduling sequence. By denoting $\alpha_t = 1 - \beta_t$ and $\bar{\alpha}_t = \prod_{s=1}^t \alpha_s$, the diffused image $\mathbf{x}_t$ at timestep $t$ has a closed form as follows:

$$\mathbf{x}_t = \sqrt{\bar{\alpha}_t}\mathbf{x}_0 + \sqrt{1-\bar{\alpha}_t}\boldsymbol{\epsilon}_t, \text{ where } \epsilon_t \sim \mathcal{N}(0, \mathbf{I}). \tag{2}$$

During the training process, a noise-predictor $\boldsymbol{\epsilon}_\theta$ learns to estimate the noise $\boldsymbol{\epsilon}$ that was previously added to $\mathbf{x}_0$ by minimizing the denoising loss:

$$L(\theta) = \mathbb{E}_{\mathbf{x}_0, \boldsymbol{\epsilon}, t}\left[\|\boldsymbol{\epsilon} - \boldsymbol{\epsilon}_\theta(\mathbf{x}_t, t)\|^2\right]. \tag{3}$$

After that, in the reverse diffusion process, a random Gaussian noise $\mathbf{x}_T \sim \mathcal{N}(\mathbf{0}, \mathbf{I})$ is iteratively denoised to reconstruct the original image $\mathbf{x}_0 \in q(\mathbf{x}_0)$. At each denoising step, using the output of the trained noise-predictor $\boldsymbol{\epsilon}_\theta$, the mean of the less noisy image $\mathbf{x}_{t-1}$ is computed as follows:

$$\boldsymbol{\mu}_t = \frac{1}{\sqrt{\alpha_t}} \left( \mathbf{x}_t - \frac{1 - \alpha_t}{\sqrt{1 - \bar{\alpha}_t}} \boldsymbol{\epsilon}_\theta(\mathbf{x}_t, t) \right). \tag{4}$$

**Membership Inference Attacks** The membership inference attacks (MIAs), introduced by Shokri et al. (2017), aim to identify whether a specific data point was part of the model's training set. Based on the threat models or the level of access attackers have, MIAs can be classified into *white-box* and *black-box* attacks. *White-box* MIAs usually utilize the internal parameters and gradients of the diffusion model to perform threshold-based attacks (Hu & Pang, 2023; Dubiński et al., 2023), gradient-based attacks (Pang et al., 2023) and proximal initialization (Kong et al., 2024). In contrast, *black-box* MIAs such as Pang & Wang (2023) target the output generated by the diffusion models without direct access to internal parameters. Studies have demonstrated that these attacks can effectively differentiate between training and non-training samples by analyzing the generated image quality (Wu et al., 2022; Matsumoto et al., 2023; Carlini et al., 2023), and the estimation errors (Duan et al., 2023). Moreover, Li et al. (2024b) recently find that fine-tuning models on small datasets can augment their vulnerability to MIAs.

**Membership Inference Defenses** Existing studies have demonstrated that overfitting in the threat models is a primary factor contributing to their vulnerability to MIAs. Consequently, various defenses have been proposed to counter MIAs, for example, by addressing overfitting, including techniques such as adversarial regularization (Hu et al., 2021), dropout (Salem et al., 2018), overconfidence reduction (Chen & Pattabiraman, 2023), and early stopping (Song & Mittal, 2021). Furthermore, differential privacy (DP) (Yeom et al., 2018; Abadi et al., 2016; Wu et al., 2019) has been widely used to mitigate MIAs by limiting the influence of any training data point on the model. However, DP methods often face trade-offs between privacy and utility. Additionally, knowledge distillation-based defenses such as distillation for membership privacy (Shejwalkar & Houmansadr, 2021) and complementary knowledge distillation (Zheng et al., 2021) aim to protect against MIAs by transferring knowledge from unprotected models. More recently, multiple techniques (Tang et al., 2022; Mazzone et al., 2022; Li et al., 2024a) have been proposed to combine knowledge distillation with ensemble learning to preserve data privacy. Nevertheless, none of the methods are designed specifically for diffusion models which are usually large and constrained by resource limitation.

**Diffusion Memorization and Mitigation** It is widely recognized that generative language models pose a risk of replicating content from their training data (Carlini et al., 2021; 2022). Similarly, Webster (2023) observe the same behavior of large diffusion models, while Somepalli et al. (2023a) argue that diffusion models trained on smaller datasets tend to produce images that closely resemble those in the training set. As the size of the training dataset increases, the likelihood of such replication decreases. Several mitigation strategies have been explored to address these issues of diffusion models by either modifying the text conditioning (Somepalli et al., 2023b; Wen et al., 2024; Ren et al., 2024), manipulating the guidance scale (Chen et al., 2024), or model pruning (Struppek et al.; Chavhan et al., 2024).

## 3 METHODOLOGY

### 3.1 MEMBERSHIP INFERENCE ATTACKS (MIAS) AND DEFENSES

Given an image $\mathbf{x}$ and a pre-trained diffusion model $\boldsymbol{\epsilon}_\theta$ on the training dataset $D_{\text{train}}$. Denoting the test dataset by $D_{\text{test}}$, the goal of MIAs (Shokri et al., 2017) is to detect if this image belongs to $D_{\text{train}}$. By viewing this as a binary classification problem, we have the dataset $B = \{(\mathbf{x}_i, y_i)\}_{i=1}^m$, where

$$y_i = \begin{cases} 1, & \text{if } \mathbf{x}_i \in D_{\text{train}} \\ 0, & \text{if } \mathbf{x}_i \notin D_{\text{train}}. \end{cases}$$

The task of MIAs then becomes to learn an attack function $\boldsymbol{f}_{\boldsymbol{\epsilon}_\theta}$ for the model $\boldsymbol{\epsilon}_\theta$ to maximize the probability of $\boldsymbol{f}_{\boldsymbol{\epsilon}_\theta}(\mathbf{x}_i) = y_i$, i.e.,

$$\max_{\boldsymbol{f}_{\boldsymbol{\epsilon}_\theta}} \mathbb{P}\left( \boldsymbol{f}_{\boldsymbol{\epsilon}_\theta}(\mathbf{x}_i) = y_i \right).$$

The design of $f_{\epsilon_\theta}$ depends on the specific choice of attacks and the attack settings. For example, in white-box attacks, the attacker can access all or parts of the training configuration and the model $\epsilon_\theta$. In black-box attacks, the attacker can only access the images generated by the model. Regardless of the settings, MIAs are usually based on the assumption that the models overfit the training data. For example, consider diffusion models, in which the model $\epsilon_\theta$ takes the input image $\mathbf{x}$, condition $\mathbf{c}$ ($\mathbf{c} = \emptyset$ in unconditional case), timestep $t \in \{1, \ldots, T\}$ and a random noise $\epsilon \sim N(\mathbf{0}, \mathbf{I})$ to compute the denoising loss in Eq. 3, we have the following assumption:

$$\mathbb{E}_{(\mathbf{x}_{\text{train}}, \mathbf{c}_{\text{train}}) \in D_{\text{train}}} \left[ \|\epsilon_\theta (\mathbf{x}_{\text{train}}, \mathbf{c}_{\text{train}}, t) - \epsilon\| \right] \leq \mathbb{E}_{(\mathbf{x}_{\text{test}}, \mathbf{c}_{\text{test}}) \in D_{\text{test}}} \left[ \|\epsilon_\theta (\mathbf{x}_{\text{test}}, \mathbf{c}_{\text{test}}, t) - \epsilon\| \right].$$

The larger the gap between the two terms, the more accessible the attacker can extract the training data. Therefore, our defense aims to make this assumption less intense so that the attacker cannot separate member data from non-member data using this property. To this aim, we design a new training paradigm to minimize the gap between train and test data, which is equivalent to the following optimization problem:

$$\min_{\epsilon_\theta} \mathbb{E}_{\substack{(\mathbf{x}_{\text{train}}, \mathbf{c}_{\text{train}}) \in D_{\text{train}} \\ (\mathbf{x}_{\text{test}}, \mathbf{c}_{\text{test}}) \in D_{\text{test}}}} \left[ \|\epsilon_\theta (\mathbf{x}_{\text{train}}, \mathbf{c}_{\text{train}}, t) - \epsilon\| - \|\epsilon_\theta (\mathbf{x}_{\text{test}}, \mathbf{c}_{\text{test}}, t) - \epsilon\| \right]. \tag{5}$$

The key idea is to modify the training loss so that our models do not fit directly into the training set. For this aim, we train two teacher models on two disjoint datasets and then let them produce "soft targets" from the other dataset to train a student model. Since these targets are the outputs of teacher models to their "non-member" images, the outputs of the student model to these data will be close to their outputs to the test data. More details are presented in the following Sections 3.2 and 3.3.

## 3.2 Disjoint Training

Although ensemble learning has been used to defend against MIAs in image classification models (Tang et al., 2022; Li et al., 2024a), they are not applicable to diffusion models, and the growing number of models poses a significant challenge when applying to large architectures. Therefore, we propose an efficient method with only two models on two disjoint subsets of the training data.

Formally, given a training dataset $D_{\text{train}}$ with no duplicated pairs of images, a test dataset $D_{\text{test}}$, and an original learning model parameterized by $\theta$. Our first step is to subdivide the training dataset into two disjoint subsets, and each is used to train a separate model, i.e., $D_{\text{train}} = D_1 \cup D_2$, where $D_1 \cap D_2 = \emptyset$. The two trained models parameterized by $\theta_1$ and $\theta_2$, respectively, can be used to generate images directly while keeping the privacy of both training subsets thanks to our customized inference pipeline proposed in Section 3.4. Alternatively, they can be distilled into a new private student model. Our basic assumption is that the two models "see" the training data of the other as test data, i.e.,

$$\mathbb{E}_{\substack{(\mathbf{x}_2, \mathbf{c}_2) \in D_2 \\ t \in \{1, \ldots, T\}}} \left[ \|\epsilon_{\theta_1} (\mathbf{x}_2, \mathbf{c}_2, t) - \epsilon\| \right] = \mathbb{E}_{\substack{(\mathbf{x}_{\text{test}}, \mathbf{c}_{\text{test}}) \in D_{\text{test}} \\ t \in \{1, \ldots, T\}}} \left[ \|\epsilon_{\theta_1} (\mathbf{x}_{\text{test}}, \mathbf{c}_{\text{test}}, t) - \epsilon\| \right].$$

$$\mathbb{E}_{\substack{(\mathbf{x}_1, \mathbf{c}_1) \in D_1 \\ t \in \{1, \ldots, T\}}} \left[ \|\epsilon_{\theta_2} (\mathbf{x}_1, \mathbf{c}_1, t) - \epsilon\| \right] = \mathbb{E}_{\substack{(\mathbf{x}_{\text{test}}, \mathbf{c}_{\text{test}}) \in D_{\text{test}} \\ t \in \{1, \ldots, T\}}} \left[ \|\epsilon_{\theta_2} (\mathbf{x}_{\text{test}}, \mathbf{c}_{\text{test}}, t) - \epsilon\| \right]. \tag{6}$$

The two models are trained with the typical denoising loss as in Eq. 3. The details of that disjoint training mechanism is presented in Algorithm 1.

## 3.3 Alternating Distillation (DistillMD)

**Choosing teacher models** Based on the assumption given in Eq. 6, we alternately use the two teacher models to generate targets for the student model to learn from. Specifically, the first model $\theta_1$, which is trained on the first subset $D_1$, will infer on the second subset $D_2$, while the second model $\theta_2$ trained on $D_2$ will infer on the first subset $D_1$. Fig. 1 illustrates the training pipeline, and the algorithm is described in Algorithm 2.

---

**Algorithm 1** Disjoint Training with DDPM

---

**Require:** Training dataset $D_{\text{train}}$, number of time steps $T$, learning rate $\eta$
1: Divide $D_{\text{train}}$ into disjoint subsets $D_1$ and $D_2$
2: Initialize two networks $\epsilon_{\theta_1}$ and $\epsilon_{\theta_2}$ with parameters $\theta_1$, $\theta_2$
3: **for** $i = 1, 2$ **do**
4:    **for** number of training iterations **do**
5:       Take sample $\mathbf{x}_0 \sim D_i$    // Sample a data point from the corresponding data distribution
6:       Sample $t \sim \text{Uniform}(\{1, \ldots, T\})$         // Randomly choose a time step
7:       Sample $\epsilon \sim \mathcal{N}(\mathbf{0}, \mathbf{I})$         // Sample noise from a Gaussian
8:       Compute $\mathbf{x}_t = \sqrt{\alpha_t}\mathbf{x}_0 + \sqrt{1 - \alpha_t}\epsilon$         // Diffuse data at time step $t$
9:       Compute loss: $L = \|\epsilon - \epsilon_{\theta_i}(\mathbf{x}_t, t)\|^2$         // Noise prediction loss
10:      Update model parameters: $\theta_i \leftarrow \theta_i - \eta\nabla_{\theta_i}L$
11:    **end for**
12: **end for**
13: **return** $\epsilon_{\theta_1}, \epsilon_{\theta_2}$

---

**Algorithm 2** Alternating Distillation (DistillMD)

---

**Require:** Disjoint data subsets $D_1$ and $D_2$, denoising networks $\epsilon_{\theta_1}$ and $\epsilon_{\theta_2}$, number of time steps $T$, number of distillation iterations $n$, learning rate $\eta$
1: Initialize student model $\epsilon_\theta$ with parameters $\theta_s$
2: **for** $i \in n$ **do**
3:    **if** $i$ is even **then**
4:       Sample $\mathbf{x}_0 \sim D_1$         // Sample a data point from the first subset
5:       $\epsilon_{\text{teacher}} = \epsilon_{\theta_2}$         // Take the second model as the teacher
6:    **else**
7:       Sample $\mathbf{x}_0 \sim D_2$         // Sample a data point from the second subset
8:       $\epsilon_{\text{teacher}} = \epsilon_{\theta_1}$         // Take the first model as the teacher
9:    **end if**
10:   Sample $t \sim \text{Uniform}(\{1, \ldots, T\})$         // Randomly choose a time step
11:   Sample $\epsilon \sim \mathcal{N}(\mathbf{0}, \mathbf{I})$         // Sample noise from a Gaussian
12:   Compute $\mathbf{x}_t = \sqrt{\alpha_t}\mathbf{x}_0 + \sqrt{1 - \alpha_t}\epsilon$         // Diffuse data at time step $t$
13:   Compute loss: $L = \|\text{stopgrad}(\epsilon_{\text{teacher}}(\mathbf{x}_t, t)) - \epsilon_{\theta_s}(\mathbf{x}_t, t)\|^2$    // Distillation loss
14:   Update model parameters: $\theta_s \leftarrow \theta_s - \eta\nabla_{\theta_s}L$
15: **end for**
16: **return** $\epsilon_{\theta_s}$

---

**Distillation loss**   To prevent the student model from overfitting to training data, the real noise term in Equation 3 is replaced by outputs of the teacher models as in Equation 7.

$$L(\theta) = \mathbb{E}_{\mathbf{x}_0, t}\left[\|\text{stopgrad}(\epsilon_{\text{teacher}}(\mathbf{x}_t, t)) - \epsilon_\theta(\mathbf{x}_t, t)\|^2\right]. \tag{7}$$

By minimizing the loss in Eq. 7 with suitable choices of the teacher models, we can make the outputs of the student model on train data closer to its outputs on test data. This closes the gap in Eq. 5 thanks to the assumption provided in Eq. 6.

In practice, our defense method can effectively mitigate both white-box and black-box attacks while maximally preserving the generation capability of the model, as shown in Section 4.

### 3.4   Self-correcting Inference Pipeline (DualMD)

**Motivation**   Black-box MIAs typically rely on training shadow models or assessing the distance between the target image and generated samples, exploiting the model's tendency to generate images close to its training data due to overfitting. However, our training paradigm ensures that for any given sample, there always exists a model that treats it as a test sample, enabling uniformly diverse sample generation.

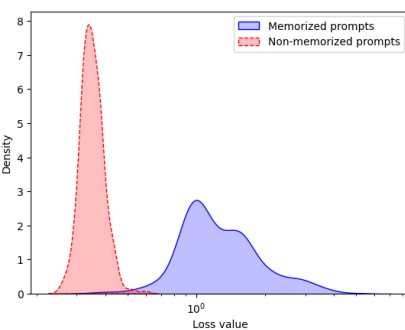

Figure 3: Distribution of the values of the loss in Eq. 8 between 500 memorized prompts and 500 non-memorized prompts on Stable Diffusion v1.5 (Rombach et al., 2022).

Table 1: Mitigation results of our methods. **Bold** and underlined numbers are the best and the second best, respectively.

| Method | SSCD ($\downarrow$) | CLIP Scores ($\uparrow$) |
|---|---|---|
| No mitigation | 0.60 | 0.26 |
| Wen et al. (2024) | 0.28 | 0.25 |
| DualMD | 0.52 | 0.27 |
| DistillMD | **0.27** | **0.28** |

Diffusion models uniquely require an iterative inference process that run the model multiple times. We leverage this characteristic by using our two teacher models to "correct" each other during inference. For instance, if the noisy image at time step $t$ causes model 1 to produce output close to the target image, model 2 will generate a more uniformly distributed image at time step $t - 1$. This "self-correcting" inference process ensures diverse generation instead of concentration near training samples. Our experimental results in Section 4 demonstrate that this method efficiently mitigates black-box MIAs on text-to-image diffusion models.

### 3.5 ENHANCING PRIVACY FOR CONDITIONAL DIFFUSION MODELS

Although disjoint training divides the data into disjoint subsets, text prompts in different subsets can still have overlapping words or textual styles that can be overfitted by both models. Therefore, we propose to enhance prompt diversity during training by using an image conditioning model to generate multiple prompts for each image of the training dataset. Then, a prompt is randomly sampled for each image in each epoch during training. More details about the limitations of DualMD and DistillMD on text-to-image diffusion models and the significance of prompt diversification are presented in Section 4.2.

### 3.6 MEMBERSHIP INFERENCE DEFENSES HELP MITIGATING DATA MEMORIZATION

Recently, an increasing body of research (Somepalli et al., 2023a;b) has highlighted the issue of data memorization in modern diffusion models, where some generated images are near-identical reproductions of images from the training datasets. Previous studies (Shokri et al., 2017; Yeom et al., 2018) have shown that overfitting renders models vulnerable to MIAs. Given that data memorization is often considered a more extreme form of overfitting, this raises an important question: *Is there a connection between MIAs and data memorization?*

Our findings suggest that loss-based MIA techniques can effectively detect memorization in diffusion models. Specifically, we use the *t-error* (Eq. 8) introduced by Duan et al. (2023) to detect whether the model memorizes a prompt. This detection is applied to a set of 500 memorized prompts and 500 non-memorized prompts (Wen et al., 2024). The resulting detection performance is reported in Fig. 3, where it is evident that the loss function effectively distinguishes between memorized and non-memorized prompts. Given this observed link between data memorization and MIAs, we are led to explore a further question: *Can membership inference defenses help mitigate data memorization?*

To investigate this, we conduct experiments on Stable Diffusion v1.5 (Rombach et al., 2022) as detailed in Section 4.3. More information about the *t-error* and the memorization experiments are presented in Appendix A.1.

Table 2: Quantitative evaluation of the quality of the defended models compared to the original model. Unconditional diffusion model is evaluated on CIFAR10 with DDPM, and text-to-image diffusion model is evaluated on Pokemon and Naruto datasets with SDv1.5. **Bold** and underlined numbers are the best and the second best, respectively.

| | **CIFAR10** | | **Pokemon** | | **Naruto** | |
|---|---|---|---|---|---|---|
| **Method** | FID ($\downarrow$) | IS ($\uparrow$) | FID ($\downarrow$) | IS ($\uparrow$) | FID ($\downarrow$) | IS ($\uparrow$) |
| Original model | **14.127** | **8.586** | **0.22** | 3.02 | **0.18** | 2.16 |
| DualMD | 21.389 | 8.011 | 0.26 | 3.34 | **0.18** | 2.12 |
| DistillMD | 14.192 | 8.391 | 0.44 | **3.52** | 0.22 | **2.19** |

# 4 EXPERIMENTS

We present the effectiveness of our defenses against white-box MIAs in Section 4.1 and against black-box MIAs in Section 4.2. We also analyze the importance of prompt diversification and find that this technique significantly enhances defense in black-box case. Ablation studies on adaptive attacks and distillation algorithms are provided in the Appendix.

**Datasets** We utilize various datasets to verify the effectiveness of the methods. The unconditional experiments use CIFAR10 (Krizhevsky et al., 2009), CIFAR100 (Krizhevsky et al., 2009), Tiny-ImageNet (Le & Yang), and STL10-Unlabeled (Coates et al., 2011) datasets. For the text-to-image experiments, we employ the popular Pokemon [1] and Naruto[2] datasets. Each dataset is divided equally, with one half used for training the models (member set) and the other half serving as the non-member set. For the model, we fine-tune the Stable Diffusion v1.5 (SDv1.5) [3] (Rombach et al., 2022) and the Stable Diffusion v2.1 (SDv2.1) [4] (Rombach et al., 2022) on Pokemon and Naruto datasets so that it overfits to the dataset. We train the default DDPM (Ho et al., 2020) from scratch for other datasets. More training details are given in Appendix A.2.

**Metrics** Following Kong et al. (2024), we utilize Area Under the ROC Curve (AUC) and True Positive Rate when the False Positive Rate is $1\%$ (TPR@1%FPR) as the key metrics to measure the vulnerability of the models to MIAs. Since we are defending MIAs, an AUC closer to 0.5 indicates better performance. For quality measurements, the popular Frenchet Inception Distance (FID) and Inception Score (IS) are measured. For unconditional models, FID and IS are computed on 25,000 generated images. In contrast, for text-to-image models, these metrics are calculated on images generated from training prompts.

Table 2 presents the quantitative performance of our methods in terms of quality preservation compared to the baseline model. It can be seen that DistillMD shows superior quality preservation in unconditional models, whereas DualMD performs better for conditional models. Additional quantitative and qualitative results are provided in Appendices A.3 and A.7, respectively. Moreover, we later observe a similar trend in defending against MIAs, which indicates that DualMD is more effective for text-to-image diffusion models, while DistillMD is better suited for unconditional diffusion models.

## 4.1 WHITE-BOX ATTACKS

For white-box MIAs, we perform two attacks SecMIA (Duan et al., 2023) and PIA (Kong et al., 2024) and defend against them with DistillMD. Since the attackers are assumed to have white-box access to the model, it is not realistic to perform DualMD defense. The results for unconditional diffusion models are given in Table 3, and for text-to-image diffusion models in Table 4. Although

---

[1] https://huggingface.co/datasets/lambdalabs/pokemon-blip-captions
[2] https://huggingface.co/datasets/lambdalabs/naruto-blip-captions
[3] https://huggingface.co/stable-diffusion-v1-5/stable-diffusion-v1-5
[4] https://huggingface.co/stabilityai/stable-diffusion-2-1-base

Table 3: Effectiveness of our DistillMD against white-box MIAs on DDPM. The closer AUC to 0.5, the better. **Bold** numbers are the best.

| | | CIFAR10 | | CIFAR100 | | Tiny-ImageNet | | STL10-Unlabeled | |
|---|---|---|---|---|---|---|---|---|---|
| **Attack** | **Method** | AUC | TPR@1% FPR (↓) | AUC | TPR@1% FPR (↓) | AUC | TPR@1% FPR (↓) | AUC | TPR@1% FPR (↓) |
| SecMIA | No defense | 0.93 | 0.35 | 0.96 | 0.45 | 0.96 | 0.53 | 0.94 | 0.30 |
| | DistillMD | **0.59** | **0.03** | **0.61** | **0.02** | **0.57** | **0.02** | **0.58** | **0.02** |
| PIA | No defense | 0.89 | 0.13 | 0.88 | 0.14 | 0.84 | 0.08 | 0.83 | 0.09 |
| | DistillMD | **0.59** | **0.02** | **0.59** | **0.03** | **0.56** | **0.02** | **0.58** | **0.02** |

Table 4: Effectiveness of our DistillMD without prompt diversification against white-box MIAs on SDv1.5 and SDv2.1. The closer AUC to 0.5, the better. **Bold** numbers are the best.

| Dataset | Attack | Method | SDv1.5 | | SDv2.1 | |
|---|---|---|---|---|---|---|
| | | | AUC | TPR@1%FPR (↓) | AUC | TPR@1%FPR (↓) |
| Pokemon | SecMIA | No defense | 0.99 | 0.79 | 0.98 | 0.189 |
| | | DistillMD | **0.48** | **0.02** | **0.54** | **0.019** |
| | PIA | No defense | 0.46 | 0.02 | 0.45 | 0.012 |
| | | DistillMD | **0.49** | **0.01** | **0.47** | **0.007** |
| Naruto | SecMIA | No defense | 0.93 | 0.475 | 0.90 | 0.333 |
| | | DistillMD | **0.46** | **0.005** | **0.45** | **0.006** |
| | PIA | No defense | 0.45 | 0.007 | 0.47 | **0.008** |
| | | DistillMD | **0.48** | **0.006** | **0.48** | **0.008** |

PIA cannot attack fine-tuned Stable Diffusion model, it is still clear that DistillMD significantly increases the privacy of both unconditional diffusion models and text-to-image diffusion models.

## 4.2 BLACK-BOX ATTACKS

For black-box MIAs, we employ the recently proposed attack in Pang & Wang (2023), which utilizes text guidance to augment the attack. The SDv1.5 model is fine-tuned on the Pokemon dataset with and without our methods. The results in Table 5 show that training defenses alone cannot completely defend against MIAs. To address this, we introduce prompt diversification training, utilizing the BLIP model (Li et al., 2022) to generate five additional prompts for each image. During training, one prompt is randomly drawn from the six (including the original) to serve as the text condition for the image. Both DistillMD and DualMD significantly mitigate MIAs with prompt diversification, highlighting the importance of prompt overfitting. Moreover, DualMD can not only better preserve the generation quality but also better defend in the case of text-to-image diffusion models.

Table 5: Effectiveness of our defenses against black-box MIA on SDv1.5. The closer AUC to 0.5, the better. **Bold** and underlined numbers are the best and the second best, respectively.

| | w/o prompt diversification | | w/ prompt diversification | |
|---|---|---|---|---|
| **Method** | AUC | TPR@1%FPR (↓) | AUC | TPR@1%FPR (↓) |
| No defense | 0.90 | 0.57 | 0.45 | 0.009 |
| DualMD | 0.82 | 0.35 | **0.52** | 0.014 |
| DistillMD | **0.66** | **0.09** | 0.46 | **0.005** |

## 4.3 MEMBERSHIP INFERENCE DEFENSES MITIGATE DATA MEMORIZATION

We use the fine-tuned SDv1.5 model using our methods to evaluate its capability of data memorization. For comparison, we employ the inference-time memorization mitigation method proposed by

Wen et al. (2024), which reduces memorization by adjusting the prompt embedding to minimize the difference between unconditional and text-conditional noise predictions.

To measure the level of memorization, we calculate the SSCD similarity score (Pizzi et al., 2022; Somepalli et al., 2023b) between the generated images and the images in the training dataset, given the same set of prompts. In addition, the CLIP score (Radford et al., 2021) is used to assess the alignment between the generated images and their corresponding prompts. A lower SSCD similarity score indicates reduced memorization, while a higher CLIP score reflects better alignment between the generated image and the prompt.

**Results and Discussion**   Table 1 shows the effectiveness of our proposed method in mitigating data memorization. A thoroughly fine-tuned model without any mitigation produces highly similar images with an SSCD similarity score of 0.60 for given prompts, indicating significant memorization. In contrast, our DualMD and DistillMD approaches significantly reduce the SSCD score to 0.52 and 0.27, respectively, suggesting that membership inference defenses can help mitigate data memorization. Notably, both methods also show a slight improvement in CLIP scores. Furthermore, the method proposed by Wen et al. (2024), which directly targets mitigating memorization, achieves an SSCD similarity score of 0.28. Our DistillMD approach, despite being designed to defend against MIAs, not only reduces data memorization more effectively but also improves image-text alignment compared to the most recently proposed method in Wen et al. (2024).

## 5 CONCLUSION

This paper presents comprehensive and novel approaches to protect diffusion models against training data leakage while mitigating model memorization. Our methodology focuses on training two models using disjoint subsets of the training data. This results in two significant contributions, including DualMD for private inference and DistillMD for developing a privacy-enhanced student model. Both techniques effectively reduce model overfitting to training samples. We further enhance privacy protection for text-conditioned diffusion models by diversifying training prompts, preventing models from overfitting specific textual patterns. Notably, our experiments reveal that model memorization represents a more severe form of overfitting than membership inference attacks (MIAs), and our unified approach successfully addresses both vulnerabilities simultaneously, eliminating the need for separate mitigation strategies. In short, our paper presents inference-time and training-time strategies to defend diffusion models against MIAs. It provides new insights into the intersection between MIAs and model memorization, advancing our understanding of privacy preservation in generative models.

**Limitations and Future Directions**   Our methods rely on dividing the training dataset into two halves, which may limit the generative capabilities of the teacher models in data-scarce scenarios. This limitation can affect the quality of the distilled model, as evidenced by the slight performance degradation shown in Table 2. Future research could focus on developing methods that allow models to leverage the entire dataset during training while maintaining strong privacy guarantees, potentially enhancing the performance of all models.

Furthermore, our inference-time defense method (DualMD) requires storing and alternating between two models, which may limit its applicability in resource-constrained environments. Future work could explore inference-time solutions that, like our DistillMD method, do not necessitate additional model storage while maintaining robust privacy protection.

## 6 REPRODUCIBILITY STATEMENT

We provide comprehensive details of all hyperparameters and experimental settings in Section 4 and Appendices A.1 and A.2. Our implementation code, included in the supplementary materials, contains clear instructions for reproduction. All models and datasets employed in our study are publicly accessible.

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

## A  APPENDIX

### A.1  SECMI LOSS

Diffusion models optimize the variational bound $p_\theta(\mathbf{x}_0)$ by matching the forward process posteriors at each step $t$. The local estimation error for a data point $\mathbf{x}_0$ at time $t$ is then expressed as:

$$\ell_{t,\mathbf{x}_0} = ||\hat{\mathbf{x}}_{t-1} - \mathbf{x}_{t-1}||^2,$$

where $\mathbf{x}_{t-1} \sim q(\mathbf{x}_{t-1}|\mathbf{x}_t, x_0)$ and $\hat{\mathbf{x}}_{t-1} \sim p_\theta(\mathbf{x}_{t-1}|\mathbf{x}_t)$. Due to the non-deterministic nature of the diffusion and denoising processes, calculating this directly is intractable. Instead, deterministic processes are used to approximate these errors:

$$\mathbf{x}_{t+1} = \phi_\theta(\mathbf{x}_t, t) = \sqrt{\bar{\alpha}_{t+1}} f_\theta(\mathbf{x}_t, t) + \sqrt{1 - \bar{\alpha}_{t+1}} \epsilon_\theta(\mathbf{x}_t, t),$$
$$\mathbf{x}_{t-1} = \psi_\theta(\mathbf{x}_t, t) = \sqrt{\bar{\alpha}_{t-1}} f_\theta(\mathbf{x}_t, t) + \sqrt{1 - \bar{\alpha}_{t-1}} \epsilon_\theta(\mathbf{x}_t, t),$$

where $f_\theta(\mathbf{x}_t, t) = \frac{\mathbf{x}_t - \sqrt{1 - \bar{\alpha}_t} \epsilon_\theta(\mathbf{x}_t, t)}{\sqrt{\bar{\alpha}_t}}$. Define $\Phi_\theta(\mathbf{x}_s, t)$ as the deterministic reverse and $\Psi_\theta(\mathbf{x}_t, s)$ as the deterministic denoise process:

$$\mathbf{x}_t = \Phi_\theta(\mathbf{x}_s, t) = \phi_\theta(\cdots \phi_\theta(\phi_\theta(\mathbf{x}_s, s), s+1), t-1)$$
$$\mathbf{x}_s = \Psi_\theta(\mathbf{x}_t, s) = \psi_\theta(\cdots \psi_\theta(\psi_\theta(\mathbf{x}_t, t), t-1), s+1)$$

Duan et al. (2023) define SecMI loss or *t-error* as the approximated posterior estimation error at step $t$:

$$\tilde{\ell}_{t,\mathbf{x}_0} = ||\psi_\theta(\phi_\theta(\tilde{\mathbf{x}}_t, t), t) - \tilde{\mathbf{x}}_t||^2, \tag{8}$$

given sample $\mathbf{x}_0 \in D$ and the deterministic reverse result $\tilde{\mathbf{x}}_t = \Phi_\theta(\mathbf{x}_0, t)$ at timestep $t$.

This SecMI loss helps identify memberships as member samples tend to have lower *t-errors* compared to hold-out samples. We leverage this *t-error* to separate memorized and non-memorized prompts. The experiment is performed similar to Wen et al. (2024) in which we plot the distribution of the loss values of the member set and the hold-out set. We utilize 500 memorized prompts of Stable Diffusion v1 extracted by Webster (2023) for the member set, and 500 non-memorized prompts that are randomly sampled from the Lexica.art prompt set [5] for the hold-out set. The result is illustrated in Fig. 3.

### A.2  TRAINING DETAILS

#### A.2.1  DATASET

Table 6 provides a summary of the diffusion models used, the datasets, and the details of the data splits.

Table 6: Adopted diffusion models and datasets.

| Model | Dataset | Resolution | # Train | # Test | Condition |
|---|---|---|---|---|---|
| DDPM | CIFAR10 | 32 | 25,000 | 25,000 | - |
| | CIFAR100 | 32 | 25,000 | 25,000 | - |
| | STL10-Unlabeled | 32 | 50,000 | 50,000 | - |
| | Tiny-ImageNet | 32 | 50,000 | 50,000 | - |
| SDv1.5 and SDv2.1 | Pokemon | 512 | 416 | 417 | text |
| | Naruto | 512 | 610 | 611 | text |

---

[5] https://huggingface.co/datasets/Gustavosta/Stable-Diffusion-Prompts

Table 7: Quantitative evaluation of the quality of the defended models compared to the original model. The evaluation utilizes the Pokemon and Naruto datasets with SDv2.1. **Bold** and underlined numbers are the best and the second best, respectively.

| Method | Pokemon | | Naruto | |
|---|---|---|---|---|
| | FID ($\downarrow$) | IS ($\uparrow$) | FID ($\downarrow$) | IS ($\uparrow$) |
| Original model | 0.44 | 2.99 | **0.16** | 2.30 |
| DualMD | **0.41** | 3.41 | 0.18 | **2.55** |
| DistillMD | **0.41** | **3.55** | 0.20 | 2.35 |

Table 8: Effectiveness of our DistillMD combining with prompt diversification against white-box MIAs on SDv1.5. The closer AUC to 0.5, the better. **Bold** numbers are the best.

| Attack | Method | w/o prompt diversification | | w/ prompt diversification | |
|---|---|---|---|---|---|
| | | AUC | TPR@1%FPR ($\downarrow$) | AUC | TPR@1%FPR ($\downarrow$) |
| SecMIA | No defense | 0.99 | 0.79 | 0.99 | 1.00 |
| | DistillMD | **0.48** | **0.02** | **0.44** | **0.01** |
| PIA | No defense | 0.46 | 0.02 | 0.61 | 0.03 |
| | DistillMD | **0.49** | **0.01** | **0.50** | **0.02** |

### A.2.2 TRAINING AND ATTACK HYPERPARAMETERS

According to Matsumoto et al. (2023), the vulnerability of the models to MIAs increases with the number of training steps because overfitting makes the models more susceptible to attacks. Therefore, in order to ensure a fair comparison, we train both the baseline model, the two models trained on two disjoint subsets, and the distilled model with the same number of training steps.

For unconditional diffusion models, we train all the models for 780,000 iterations with a batch size of 128, a learning rate of 2e-4.

For SDv1.5 and SDv2.1, we use the Huggingface Diffusers codebase [6] to fine-tune the model in 20,000 iterations, with batch size of 16 and learning rate of 1e-5.

For white-box attacks on all models, we use the codebase and default settings of SecMIA [7] (Duan et al., 2023) and PIA [8] (Kong et al., 2024)

For black-box attacks on SDv1.5, we generate 3 images for each prompt, each is generated using DDIM (Song et al., 2021) with 50 inference steps.

For evaluating data memorization in Section 4.3, we use the codebase from (Wen et al., 2024) [9].

### A.3 ADDITIONAL QUANTITATIVE RESULTS

Table 7 presents the additional quantitative performance of our methods, highlighting quality preservation compared to the baseline model with the SDv2.1 backbone.

Unlike black-box attacks discussed in Section 4.2, prompt diversification training shows only a slight improvement in defense against white-box attacks, as presented in Table 8.

---

[6]https://github.com/huggingface/diffusers/blob/main/examples/text_to_image/README.md

[7]https://github.com/jinhaoduan/SecMI

[8]https://github.com/kong13661/PIA

[9]https://github.com/YuxinWenRick/diffusion_memorization

## A.4 ATTACK ANALYSIS

To further understand our defense capability, we provide the ROC curves for various configurations under black-box MIAs in Fig. 4. While the original model is severely vulnerable to MIAs, it is evidenced that our defenses can effectively mitigate this risk, even in worst-case scenarios when the FPR is very low.

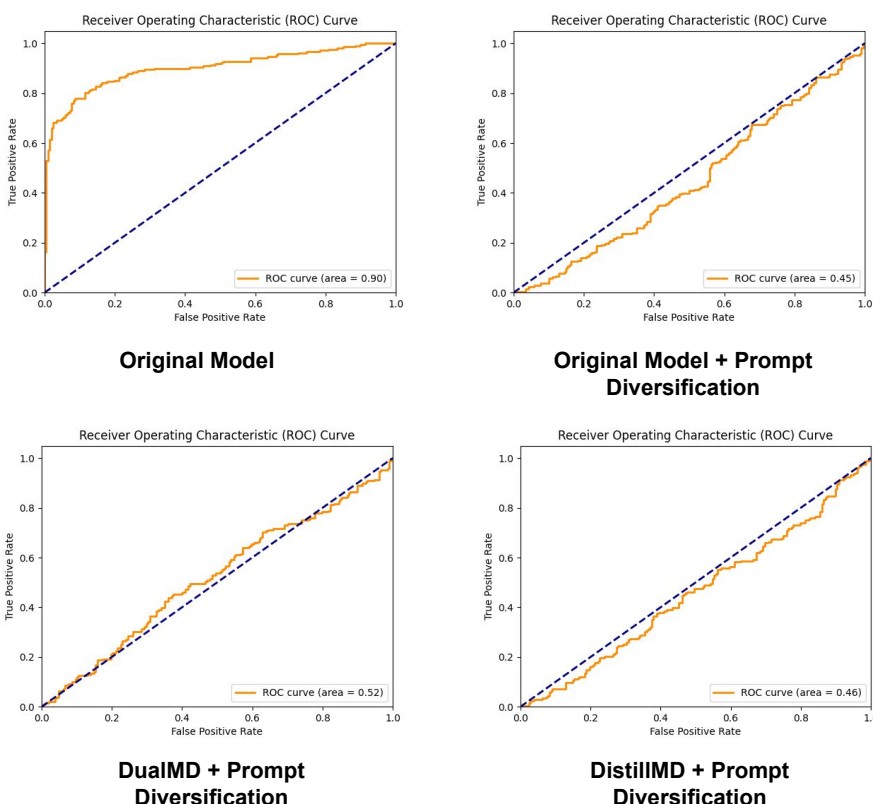

Figure 4: ROC curves of black-box MIAs comparing the originally trained model with our defense methods.

## A.5 ADAPTIVE ATTACK

To extend the robustness evaluation of our defense mechanism, we investigate its vulnerability to adaptive attacks where adversaries have complete knowledge of the defense strategy. We design an iterative attack targeting DualMD's dual-model architecture by manipulating the denoising process across multiple generation rounds. The attack proceeds as follows: First, we generate an image using $n$ denoising steps, alternating between Sub-Model1 (SB1) and Sub-Model2 (SB2). We then introduce noise at the second-to-last timestep, effectively nullifying all denoising steps except the initial one performed by SB1. This noisy image and its corresponding timestep serve as the starting point for a subsequent generation round with $n-1$ steps, beginning with SB1. By iteratively repeating this process, we systematically reduce the influence of SB2 while preserving SB1's denoising effects. When combined with black-box MIAs, this approach provides a comprehensive evaluation of our defense mechanism. As shown in Table 9, DualMD maintains its defensive efficacy even after multiple rounds of this adaptive attack on the Pokemon dataset, with an AUC remaining close to $0.5$ and very low TPR at 1% FPR.

Table 9: Performance of DualMD against our designed adaptive attack.

| Number of generation rounds | AUC | TPR@1%FPR (↓) |
|---|---|---|
| 2 | 0.53 | 0.024 |
| 3 | 0.51 | 0.048 |

## A.6 DISTILLATION ANALYSIS

Knowledge distillation has emerged as a prominent approach for mitigating Membership Inference Attacks (MIAs) in classification models (Shejwalkar & Houmansadr, 2021; Zheng et al., 2021; Tang et al., 2022; Mazzone et al., 2022; Li et al., 2024a). To demonstrate the advantages of our dual-model architecture in DistillMD, we conduct a comparative analysis against conventional knowledge distillation under SecMI attack Duan et al. (2023) using the CIFAR10 dataset. The key distinction lies in the training methodology. In particular, traditional knowledge distillation employs a single teacher model trained on the complete dataset, whereas DistillMD leverages two specialized teacher models, each trained on mutually exclusive subsets of the training data.

The experimental results presented in Table 10 reveal significant differences in defense efficacy. Although conventional knowledge distillation provides modest protection, reducing the AUC from 0.93 (no defense) to 0.74, this improvement falls short of the robustness required for real-world applications. These findings underscore the crucial role of our dataset partitioning strategy and dual-teacher architecture in DistillMD. Notably, existing distillation-based defense mechanisms for classification models often incorporate supplementary techniques, such as confidence-based sample selection Shejwalkar & Houmansadr (2021), to enhance privacy guarantees. Although these techniques have proven effective in classification scenarios, their direct application to generative models presents unique challenges. Our work establishes a foundation for future research to bridge this gap and adapt these distillation methods for diffusion models while maintaining their privacy-preserving properties.

Table 10: Performance of DualMD against our designed adaptive attack. **Bold** and underlined numbers are the best and the second best, respectively.

| Methods | AUC | TPR@1%FPR (↓) |
|---|---|---|
| No defense | 0.93 | 0.35 |
| Normal KD | 0.74 | 0.06 |
| DistillMD | **0.59** | **0.03** |

## A.7 QUALITATIVE RESULTS

In this section, we present images generated by trained models with and without our methods. Fig. 5 shows images generated on the CIFAR10 dataset in the unconditional setting, while Fig. 6 and Fig. 7 display images generated on the Pokemon dataset in the conditional setting.

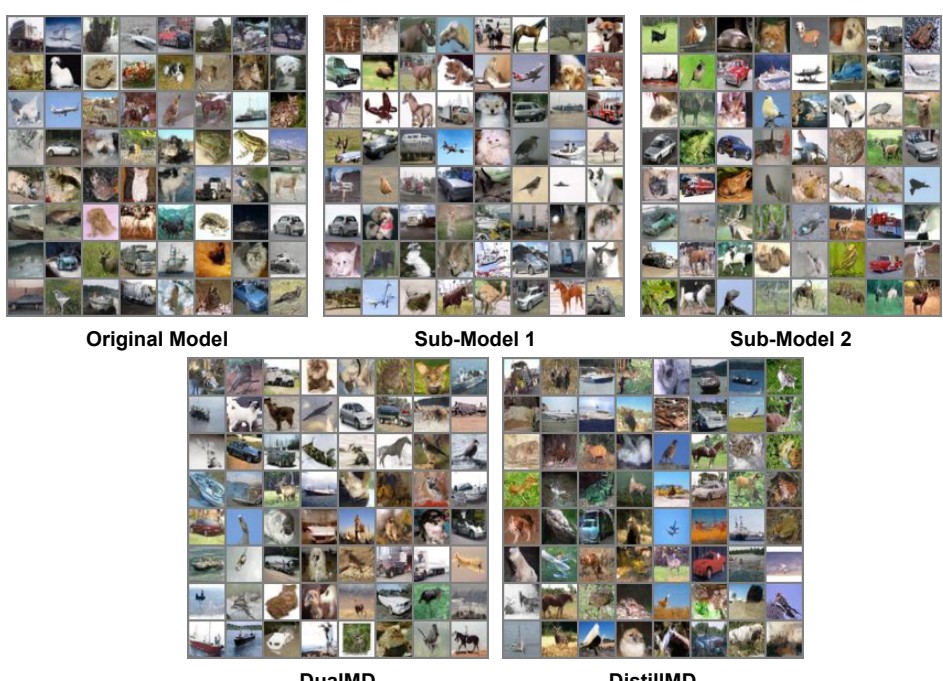

Figure 5: Images generated by models trained on CIFAR10. The Original Model was trained on the full dataset, whereas Sub-Model 1 and Sub-Model 2 were trained on two disjoint subsets. DualMD and DistillMD images were generated using our proposed methods.

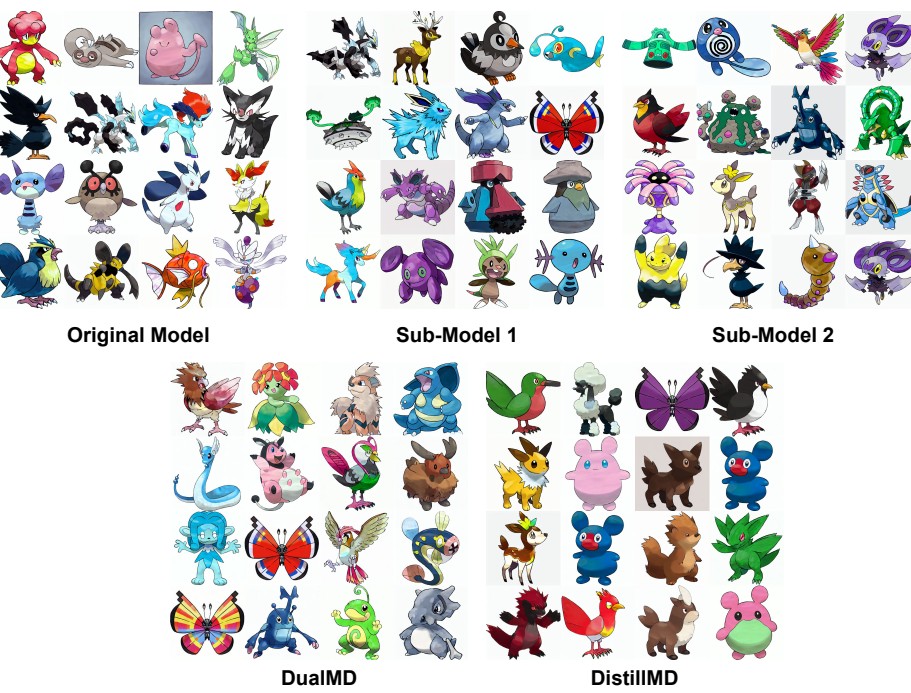

Figure 6: Images generated by models *without prompt diversification* trained on Pokemon. The Original Model was trained on the full dataset, whereas Sub-Model 1 and Sub-Model 2 were trained on two disjoint subsets. DualMD and DistillMD images were generated using our proposed methods.

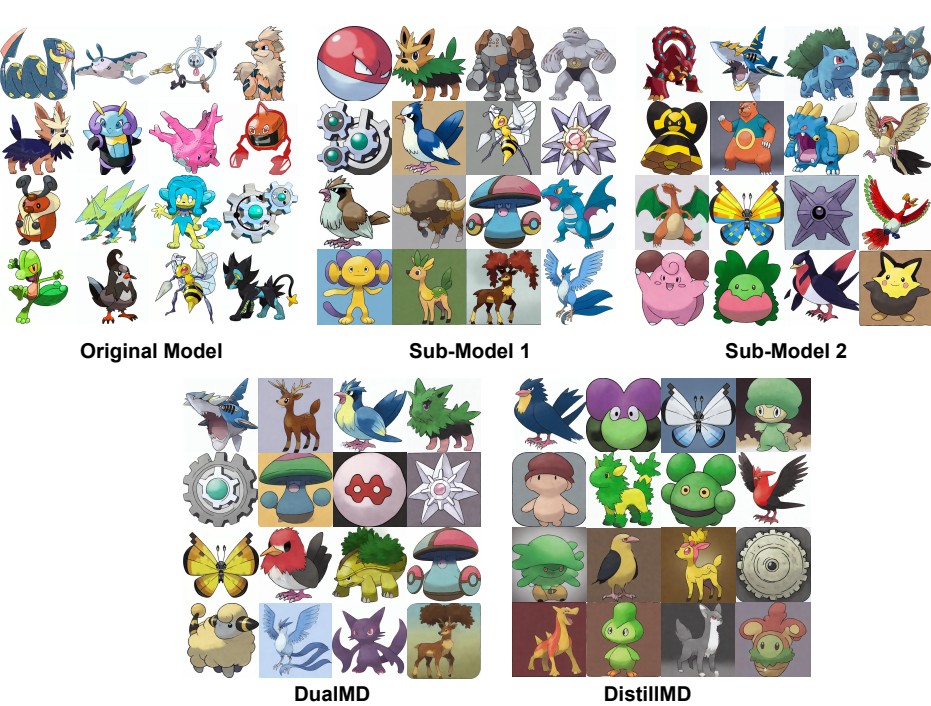

Figure 7: Images generated by models *with prompt diversification* trained on Pokemon. The Original Model was trained on the full dataset, whereas Sub-Model 1 and Sub-Model 2 were trained on two disjoint subsets. DualMD and DistillMD images were generated using our proposed methods.

