# OpenReview forum: "Dual-Model Defense: Safeguarding Diffusion Models from Membership Inference Attacks through Disjoint Data Splitting"
_ICLR.cc/2025/Conference — Submitted to ICLR 2025_

### Official Review · Reviewer_WvSX · 2024-10-27

**Soundness:** 3
**Presentation:** 2
**Contribution:** 3
**Rating:** 6
**Confidence:** 2

**Summary:**

The paper introduces two  defense strategies—DualMD and DistillMD—to protect diffusion models against Membership Inference Attacks (MIAs) without significantly compromising model utility. The two defense strategies introduced operate differently: DualMD mitigates MIA risks in black-box settings by dividing data between two models and alternating inference to reduce data exposure. DistillMD combines dual models into a student model via distillation, offering robust defenses against both white-box and black-box attacks.

**Strengths:**

DualMD and DistillMD effectively adapt ensemble learning and distillation to protect diffusion models from MIA attacks
The paper includes evaluations across various datasets, models, and attack types (both white-box and black-box). This thorough testing underscores the robustness of the proposed methods.

**Weaknesses:**

Lack of Comparative Analysis with Existing Methods: The study does not explicitly compare its defense methods with existing privacy defense methods beyond describing MIA vulnerabilities. A side-by-side comparison with traditional distillation or differential privacy approaches in image synthesis could underscore the novelty and effectiveness of the proposed methods.

Limited Discussion of Practical Constraints: The methods require dual models (DualMD) or distillation (DistillMD), both of which can be computationally intensive. Discussing potential resource constraints, such as memory limitations or model scalability in real-world applications, would make the methods' practicality more transparent.

Lack of Explanation on Evaluation Metrics: While metrics like FID, IS, and AUC are widely used, a brief explanation of their relevance to privacy defense and utility preservation would clarify the evaluation approach for readers.

Improve Clarity on Methods and Results: I find the caption for Figure 1 a bit confusing. I find the presentation a bit confusing. For example, I dont remember the DDPM being introduced but somehow where is an algorithm 1 for the DDPM while there is no algorithm for the DualMD.

**Questions:**

Could you clarify why you randomly sample t instead of doing it sequentially? in the Algorithm 1 and 2?

---

> ### Author Response · Authors · 2024-11-21
> **Response to Reviewer WvSX**
>
> We thank the reviewer for the comments, we address main concerns as follows:
>
> - __Regarding the comparison to other distillation-based method__:
> We thank the reviewer for the suggestion. We compare the performance of a normal distillation without data division with our approach in the following table. The experiment is conducted in the CIFAR10 dataset with the same settings outlined in our paper. We distill the model with a normal distillation process for 400,000 iterations. The results imply that while the method can mitigate MIAs to some extent, it is not adequate enough for practical usage.
>
> | Method         | AUC      | TPR@1%FPR ($\downarrow$) |
> | -------------- | -------- | --------- |
> | No defense     | 0.93     | 0.35      |
> | Normal Distill | 0.74     | 0.06      |
> | DistillMD      | __0.59__ | __0.03__  |
>
> - __Regarding the Practical Constraints__: Recent diffusion models [1] have scaled to billions of parameters. The Stable Diffusion 3 model can have up to 8B parameters, which makes it hard to train or infer on resource-constrained situations.
>
> - __Regarding the Evaluation Metrics__: FID and IS are widely used metrics for evaluating image quality. They help verify that the model generates high-quality images while effectively defending against MIA. The Area Under the ROC Curve (AUC) measures the capability of the attack model to classify member and non-member samples correctly, and a closer AUC to 0.5 indicates a stronger defense capability. The True Positive Rate (TPR) when the False Positive Rate (FPR) is 1% (TPR@1%FPR) measures the ability of the attack model to detect member samples correctly in worst-cases when FPR is very low.
>
> - __Regarding the explanation of Figure 1__:
> (Left): We divide the training dataset into two non-overlapping subsets. Each subset is then used to train a separate diffusion model with the vanilla diffusion loss (refer to Line 9 in Algorithm 1).
> (Right): During the distillation phase, for each iteration, if a data point belongs to subset 1, it is passed through the pre-trained model 2 (which is frozen) to generate a "soft" label. Similarly, if a data point belongs to subset 2, it is passed through the pre-trained model 1 (also frozen) to produce a soft label. The student model then uses this "soft" label as the target to compute the diffusion loss (refer to Line 13 in Algorithm 2).
>
> - __Regarding the Algorithm__: During training, we randomly sample t because we primarily rely on the vanilla diffusion loss [2] for training both models and distilling the student model.
>
> We really hope that the reviewer can reconsider the review score. Please let us know if you would like us to do anything else.
>
> Best,
>
> Authors
>
> __References:__
>
> [1] Esser, et al. Scaling rectified flow transformers for high-resolution image synthesis. International Conference on Machine Learning, 2024.
>
> [2] Ho, Jonathan, Ajay Jain, and Pieter Abbeel. "Denoising diffusion probabilistic models." Advances in neural information processing systems 33 (2020): 6840-6851.

---

### Official Review · Reviewer_sadW · 2024-11-03

**Soundness:** 2
**Presentation:** 1
**Contribution:** 2
**Rating:** 6
**Confidence:** 4

**Summary:**

The paper proposed an ensemble-like method to defend membership inference attacks (MIA) in diffusion models (DM). Instead of training a single DM on the whole dataset, the proposed method train two models on disjoint subsets of the training set. At inference time, the denoising process will be alternately done by two models. The method is tested on the SoTA MIA attacks.

**Strengths:**

* The proposed method is simple yet effective in defending MIA attacks.
* In the tested datasets and models, the proposed method is effective in defending MIA attacks.
*

**Weaknesses:**

1. The experiments lack essential baselines and ablation studies. Though most defense methods mentioned in the Related Work were not designed for diffusion models, the distillation-based method should be generally applicable. Note that the proposed method is also a distillation-based method. Thus, the distillation of a single diffusion model (following prior arts noted in the Related Work) should be included as a baseline. More detailed ablation studies should be conducted. For example, if the two models are necessary.
2. The claim in the introduction, "We evaluate the effectiveness of our methods in training large text-to-image diffusion models and propose a technique to prevent the models from overfitting to the prompts.", is misleading. In the experiment (Table 4), without using their method, the no-defense baseline can effectively reduce the risk of MIA. I don't think the proposed method contribute to the prompt overfitting.
3. The logic of the paper is confusing.
* It is not clear why the prompt overfitting is considered. In Line 112-113, the prompt overfitting is raised for model memorization, whose connection to the MIA is not clarified in the context.
* Throughout the paper, there are many forward references, making the logic very hard to follow.
* Lacks intuitive explanation on why this method works.
4. In MIA evaluation, the ROC curve should be presented to show the rigor of evaluation in MIA.
5. Only one diffusion model (SDv1.5) is used, which makes the conclusion less generalizable.

**Questions:**

* How is the proposed method compared to distillation-based baselines?

---

> ### Author Response · Authors · 2024-11-21
> **Response to Reviewer sadW**
>
> We thank the reviewer for the comments, we address main concerns as follows:
>
> - __Regarding the comparison to other distillation-based method:__ We thank the reviewer for the suggestion, and we would like to provide the performance of an ordinary distillation without data division in comparison to our approach in the following table. The experiment is conducted in the CIFAR10 dataset with the same settings outlined in our paper. We distill the model using an ordinary distillation process for 400,000 iterations. The results imply that while the method can somewhat mitigate MIAs, it is not effective enough for practical usage. It should also be noted that other distillation-based methods for safeguarding classification models also apply additional techniques, such as label smoothing, to enhance privacy. While these techniques are effective for classification models, they are not generally applicable to generative modeling tasks and we open the door for future works to employ those distillation methods on diffusion models. We hope that this analysis addresses the reviewer's concern about the importance of training multiple private models compared to distilling with only one model.
>
> | Method         | AUC      | TPR@1%FPR ($\downarrow$) |
> | -------------- | -------- | ------------------------ |
> | No defense     | 0.93     | 0.35                     |
> | Normal Distill | 0.74     | 0.06                     |
> | DistillMD      | __0.59__ | __0.03__                 |
>
> - __Regarding our claim in introduction:__ We admit that there would be a misunderstanding in the Introduction of our paper. While the prompt diversification technique can somewhat mitigate MIAs, combining it with our proposed methods can further improve the defense performance, reducing the possibilities of False Positive evidenced by a closer AUC to 0.5.
>
> - __Regarding the logic of the paper:__ We would like to provide some clarifications as follow: We consider the prompt overfitting issue for MIAs because both MIAs and model memorization are related to model overfitting and studies from model memorization revealed that diffusion models not only overfit training images but also training prompts. In addition, the reason why we train two models on two disjoint subsets is to ensure that for any training sample, there is always a model that "see" it as a test sample. Therefore, by using this model to infer the training sample, we can avoid MIAs. Our proposed methods DualMD and DistillMD further extend the use of the two models in practical situations, avoiding the need to "choose" model to perform inference. We thank you for these comments, and we will improve these points in our revised version.
>
> - __Regarding the lack of ROC curve:__ The ROC curves of different strategies under black-box MIAs will be included in our Appendix.
>
> - __Regarding more diffusion model architectures:__ We conducted experiments using the Stable Diffusion 2.1 backbone on the Pokemon dataset. The resulsts suggest that our proposed DistillMD method not only enhances image quality but also effectively defends against two attack methods: SecMIA and PIA.
>
>
> | Method         | FID ($\downarrow$) | IS ($\uparrow$) |
> |:-------------- | ------------------ | --------------- |
> | Original model | 0.44               | 2.99            |
> | DistillMD      | 0.41               | 3.55            |
>
>
> | Attack | Method     | AUC  | TPR@1%FPR ($\downarrow$) |
> |:------ | ---------- | ---- | ------------------------ |
> | SecMIA | No defense | 0.98 | 0.189                    |
> | SecMIA       | DistillMD  | 0.54 | 0.019                    |
> | PIA    | No defense | 0.45 | 0.012                    |
> | PIA       | DistillMD  | 0.47 | 0.007                    |
>
>
> We really hope that the reviewer can reconsider the review score. Please let us know if you would like us to do anything else.
>
> Best,
>
> Authors

---

> > ### Comment · Reviewer_sadW · 2024-11-22
> > **Thank you**
> >
> > Thank you for the rebuttal. My concerns are addressed.

---

> > > ### Author Response · Authors · 2024-11-23
> > > **Thank you so much for your post-rebuttal response!**
> > >
> > > Dear reviewer,
> > >
> > > We are grateful for your post-rebuttal response and score reconsideration. We are so glad that most of your concerns have been addressed.
> > >
> > > Best regards,
> > >
> > > Authors.

---

### Official Review · Reviewer_wt6x · 2024-11-07

**Soundness:** 2
**Presentation:** 2
**Contribution:** 2
**Rating:** 5
**Confidence:** 4

**Summary:**

This paper studies the membership inference defense for diffusion models by training two diffusion models on disjoint data. This work introduce two methods, DistillMD and DualMD. After training model A on dataset A and model B on dataset B, DistillMD uses model A to generate image for dataset B and model B to generate image for dataset A. DistillMD then use generated images to train a student model. DualMD use the model A and B to perform denoising together in denoising the images. At each step, DualMD use one model to perform denoising at that step, then at next step, DualMD use the other model to perform the denoising.
Experimental evaluation include unconditional diffusion model and text-to-image diffusion model, including black-box attacks and white-box attacks.

**Strengths:**

1. The idea is intuitive and easy to follow, training two diffusion models on disjoint dataset and designing models on how to design defense against MIAs based on these two models.
2. The evaluation considers both white-box attacks and black-box attacks on both unconditional diffusion models and text-to-image diffusion models.

**Weaknesses:**

I have concerns regarding the privacy and utility evaluation in this work.

1. For the privacy evaluation, it seems to me that similar to Tang et al. 2022, the disjoint dataset training in this work still face some issues for the correlated pairs, Tang et al. 2022 provides a case study how severe the correlated pairs issue is in CIFAR10 dataset. It would be helpful to investigate and discuss the correlated pairs and potential mitigations. It is also unclear to me whether DualMD is better or DistillMD is better for text-to-image diffusion models, as the AUC and TPR-FPR show different trends and Carlini et al. 2022[1] emphasize the importance of worst-case analysis for privacy leakage.

2. For the utility evaluation, it would be helpful to provide the FID and IS metric for images generated from model A and model B with the original data to understand utility trend among DualMD and DistillMD. I wonder if there is a explanation for why DualMD is better than DistillMD for text-to-image diffusion models, or utility gap is small, or the conclusion is dawn on a single dataset Pokemon that may not be true in general. Also it is unclear to me if results for Pokeman in Table 1 is with prompt diversification or without.

3. The writings could be improved. In introductions, the contribution 2 states the "proposing a technique to prevent the models from overfitting to prompts", it seems to me this method is prompt diversification and is from Somepalli et al. 2023b, maybe the author could clarify what is different from prompt diversification and Sompalli et al. 2023b.  The connection between MIA and memorization are also studied in prior works. For example, in the extraction attack for diffusion models by Carlini et al. 2023, MIA is applied to select those images that are likely to be memorized. Minor question, what is "stopgrad" in Equation (7)?


[1] Carlini et al., Membership Inference Attacks From First Principles. IEEE S&P 2022

**Questions:**

See above.

---

> ### Author Response · Authors · 2024-11-21
> **Response to Reviewer wt6x**
>
> We thank the reviewer for the comments, we address main concerns as follows:
>
> - __Regarding the correlated pairs__: While the correlated pair analysis can be valuable for classification tasks such as in [1], data deduplication can effectively search similar image pairs and remove them from training data for generative modeling tasks. Duplication retrieval or deduplication methods for diffusion model training have been widely investigated in [3][4][5] and have already been applied in large-scale model training such as in Stable Diffusion 3 [3]. Therefore, we argue that correlated training pairs is not a substantial issue for diffusion model training because it can be effectively mitigated by preprocessing the training data.
>
> - __Regarding the text-to-image task:__ We believe that DualMD better suits text-to-image diffusion models as our analysis reveals that it better preserves the generation quality while effectively mitigates MIAs. Although one limitation of DualMD is the worse performance in worst-case scenarios compared to DistillMD, a score of 1.4% still falls under a safe threshold and can be considered an effective defense.
>
> - __Regarding the utility evaluation:__ The results for the Pokemon dataset are without prompt diversification. In the table below, we additionally provide the scores of the two teacher models on CIFAR10. Although there is unavoidable performance degradation from the distillation process, this loss is offset by the significant increase in the model's privacy.
>
> | Score | Baseline | Model 1 | Model 2 | DistillMD |
> | ----- | -------- | ------- | ------- | --------- |
> | FID   | 14.13    | 13.24   | 13.59   | 14.19     |
> | IS    | 8.59     | 8.56    | 8.52    | 8.39      |
>
> - __Regarding the connection between MIAs and Memorization:__ Although prior works also investigated the connection between MIAs and Memorization, our study further crosses the border by utilizing insights from one field to the other. On the one hand, we showed that MIA losses can effectively detect memorized instances, and our proposed defense can not only defend MIAs but also mitigate memorization. On the other hand, we also revealed that insights from memorization studies, such as the prompt overfitting, can be beneficial to understanding MIAs, further improving defending mechanisms. In addition, although the propmt diversification technique is studied and applied in [6] for mitigating model memorization, we utilize it for another purpose of defending against MIAs for the first time and prove its essential role in safeguarding text-to-image diffusion models.
>
> - __Regarding the "stopgrad" notation:__ During the distillation phase, we utilize two pretrained and frozen teacher models to generate soft labels for the student model. These teacher models remain unchanged throughout this phase, so we apply the "stopgrad" operation in Equation 7 to prevent gradient updates on the teacher models.
>
> We really hope that the reviewer can reconsider the review score. Please let us know if you would like us to do anything else.
>
> Best,
>
> Authors
>
> __References:__
>
> [1] Carlini, Nicholas, et al. "Membership inference attacks from first principles." 2022 IEEE Symposium on Security and Privacy (SP). IEEE, 2022.
>
> [2] Tang, et al. "Mitigating Membership Inference Attacks by Self-Distillation Through a Novel Ensemble Architecture". Proceedings of the 31st USENIX Security Symposium.
>
> [3] Esser, et al. Scaling rectified flow transformers for high-resolution image synthesis. International Conference on Machine Learning, 2024.
>
> [4] Carlini, Nicolas, et al. "Extracting training data from diffusion models." 32nd USENIX Security Symposium (USENIX Security 23), 2023.
>
> [5] Somepalli, Gowthami, et al. "Diffusion art or digital forgery? investigating data replication in diffusion models." Proceedings of the IEEE/CVF Conference on Computer Vision and Pattern Recognition, 2023.
>
> [6] Somepalli, Gowthami, et al. "Understanding and mitigating copying in diffusion models." Advances in Neural Information Processing Systems 36, 2023.

---

> > ### Comment · Reviewer_wt6x · 2024-11-26
> >
> > Thank you for you response. I still have my concerns after reading the response. I listed my concerns as follows.
> >
> > 1. Correlated pairs. [R1] has discussed the challenge of identifying the similar pairs for the Near Access-Freeness (NAF) framework [R2] that utilized two models on two subsets for generative model copyright protection. Although copyright protection is a different goal from membership inference attacks, I think the challenge of the similar pairs still hold for MIAs. [R3] provides a detailed analyze showing the adaptive attacks for [R4] and highlights the necessity of evaluation should consider the worst case. Therefore I think such adaptive attack is necessary.
> >
> > 2. Utility evaluation. Thank you for providing the additional results for the CIFAR10. I think providing the results for text-to-image tasks would also be helpful, especially with prompt diversification.  Providing the results for with/without prompt diversification would also help understand the effect on utility with such prompt diversification. Besides, I wonder if the some intuitive explanation on why DualMD has better utility on text-to-image task and Distill has better utility on unconditioned generation.
> >
> > 3. Memorization and MIAs. As noted in the initial review, prior work like [R4] use MIAs losses to detect memorized images in diffusion model. I feel that the current manuscript has several over-claims regarding the contribution for the first to establish the connection between memorization and MIAs.
> >
> >
> >
> >
> > [R1] Chen et al. Randomization Techniques to Mitigate the Risk of Copyright Infringement. https://arxiv.org/pdf/2408.13278
> >
> > [R2] Vyas et al.  On provable copyright protection for generative models. ICML 2023. https://openreview.net/pdf?id=qRAHZVnQNY
> >
> > [R3] Aerni et al. Evaluations of Machine Learning Privacy Defenses are Misleading. CCS 2024. https://arxiv.org/pdf/2404.17399
> >
> > [R4] Tang et al. Mitigating Membership Inference Attacks by Self-Distillation Through a Novel Ensemble Architecture. USENIX 2022. https://www.usenix.org/system/files/sec22-tang.pdf
> >
> > [R5] Carlini et al. Extracting training data from diffusion models. USENIX Security 2023. https://www.usenix.org/system/files/usenixsecurity23-carlini.pdf

---

> > > ### Author Response · Authors · 2024-11-28
> > > **Response to Reviewer wt6x (Part 1)**
> > >
> > > We greatly appreciate the reviewer’s response to our rebuttal. We address the additional questions as follows.
> > >
> > > - **Regarding the Correlated Pairs:** We thank the reviewer for raising this critical point. We agree that challenges exist in identifying similar pairs in datasets, and further investigation is needed to mitigate these issues. However, this concern lies outside the scope of our current study, as our work assumes no duplicated pairs in the training data in order to have two disjoint training subsets. We will revise the manuscript to clarify this assumption. Additionally, efficient deduplication algorithms, such as the SSCD algorithm referenced in Sections E.2 and E.3 of the Stable Diffusion 3 paper [1], are widely adopted in large-scale diffusion model training, which we believe can address this challenge to some extent.
> > >
> > > - **Regarding the Adaptive Attack:** Label-based adaptive attacks commonly used for classification models are not directly applicable to generative models, especially text-to-image diffusion models. In our study, we designed a query-based adaptive black-box attack against DualMD and reported the results in Appendix A.5; please refer to this section for more detailed analysis. As detailed in our previous response, the case of adaptive attacks with duplicated canaries is unrealistic, particularly when effective deduplication techniques are applied during diffusion model training (as described in [1]). Furthermore, the SecMI [2] and PIA [3] attacks employed in our paper are state-of-the-art MIAs specifically tailored for diffusion models. Our study avoids the second pitfall of weak attacks, as discussed in [4].
> > >
> > > - **Regarding the Utility Evaluation:** We thank the reviewer for that suggestion. We agree that results reflecting the utility impact of prompt diversification would add value. Hence, we would like to provide the evaluation metrics for models with and without prompt diversification below:
> > >
> > > | Prompt Diversification | Model     | FID ($\downarrow$) | IS ($\uparrow$) |
> > > | ---------------------- | --------- | ------------------ | --------------- |
> > > | without                | Baseline  | 0.22               | 3.02            |
> > > | with                   | Baseline  | 0.45               | 3.37            |
> > > | without                | Model 1   | 0.19               | 2.85            |
> > > | with                   | Model 1   | 0.29               | 2.83            |
> > > | without                | Model 2   | 0.19               | 2.98            |
> > > | with                   | Model 2   | 0.30               | 2.96            |
> > > | without                | DualMD    | 0.26               | 3.34            |
> > > | with                   | DualMD    | 0.61               | 4.13            |
> > > | without                | DistillMD | 0.44               | 3.52            |
> > > | with                   | DistillMD | 0.98               | 3.65            |
> > >
> > >
> > > We recognize that while the FID score of the model with prompt diversification is less optimal compared to the model without prompt diversification, the IS score is comparable and even shows improvement in some cases. Furthermore, the qualitative results presented in Appendix A.7 demonstrate that the images generated with prompt diversification method exhibit a visual quality comparable to those generated without it.

---

> > > > ### Author Response · Authors · 2024-11-28
> > > > **Response to Reviewer wt6x (Part 2)**
> > > >
> > > > - **Regarding the Intuitive Explanation:** We would like to provide an explanation based on our understanding here. For text-to-image diffusion models, the efficacy of the prompt diversification technique implies that prompts play a crucial role in shaping the generated images. Hence, outputs of the two sub-models for the same prompt would be more consistent compared to the unconditional case because every denoising step is guided by the prompt condition, making it possible for DualMD to generate good images. However, in the unconditional case, there is more stochasticity in the generation process as the final output depends only on the initial noise, and the two sub-model's trajectories diverge after every denoising step. Therefore, alternately using the two models for generation would result in a more random trajectory between the two sub-models.
> > > >
> > > > - **Regarding Memorization and MIAs:** We thank the reviewer for pointing out that. We admit that our initial statement may have caused confusion. While we are not the first to leverage MIA losses for memorization detection (as demonstrated in [2]), our study is the first to establish a *bidirectional connection* between memorization and MIAs. Specifically, we utilize insights from MIAs to detect memorization and leverage insights from memorization (e.g., prompt overfitting) to enhance defenses against MIAs. Furthermore, we show that our method simultaneously mitigates both MIAs and model memorization. We will revise our claims to reflect these contributions more accurately.
> > > >
> > > >
> > > > Please let us know if our revisions adequately address your concerns. Your further opinions are significant for evaluating our revised paper and we hope to hear from you soon. Thank you again for your effort and constructive suggestions.
> > > >
> > > > Best regards,
> > > >
> > > > The Authors.
> > > >
> > > > **References:**
> > > >
> > > > [1] Esser, et al. Scaling rectified flow transformers for high-resolution image synthesis. International Conference on Machine Learning, 2024.
> > > >
> > > > [2] Duan, Jinhao, et al. Are diffusion models vulnerable to membership inference attacks?. International Conference on Machine Learning. PMLR, 2023.
> > > >
> > > > [3] Kong, Fei, et al. An Efficient Membership Inference Attack for the Diffusion Model by Proximal Initialization. The Twelfth International Conference on Learning Representations.
> > > >
> > > > [4] Aerni, Michael, Jie Zhang, and Florian Tramèr. Evaluations of Machine Learning Privacy Defenses are Misleading. arXiv preprint arXiv:2404.17399 (2024).

---

> > > > > ### Author Response · Authors · 2024-12-03
> > > > > **Follow-up: Have our clarifications addressed the additional concerns?**
> > > > >
> > > > > Dear Reviewer wt6x,
> > > > >
> > > > > Once again, thank you very much for your efforts in reviewing this paper. We have tried our best to address all your additional concerns and provided clarifications on all confusing concepts.
> > > > >
> > > > > As the discussion period is nearing an end (in a few hours), we would like to know if you can review our newest replies to see if they successfully answered your questions. This also gives us a decent amount of time to address any of your remaining/additional concerns. We would greatly appreciate it if you could reconsider the score of our manuscript.
> > > > >
> > > > > Best regards,
> > > > >
> > > > > Authors.

---

> > > > > > ### Comment · Reviewer_wt6x · 2024-12-03
> > > > > >
> > > > > > Thank you for your response. I have read your follow-up response and decided to maintain my score.
> > > > > >
> > > > > > > Correlated pairs and adaptive attacks.
> > > > > >
> > > > > > Indeed, [R3] and [R4] focus on the similar pairs but with different labels. I am aware that the diffusion model could also use the class label as a precondition by adding a class label embedding for image generation[R6]. I also want to clarify the difference for correlated pair setting between [R3] and [R4] (and I apologize for not making this clear earlier due to my conflict of time, though [R4] states the setting). [R3] use the correlated pairs with different labels in the training set to emphasize the significance the privacy leakage , [R4] analyze such correlated pairs with different labels between training set and test set. It seems to me here, for such class label-based diffusion models, sample A in training set and sample B in test set where A and B are semantic similar but with different labels, would have similar issue in the correlated pair attack, as the MIA statistics computed by these two samples would favor one specific class over the other, and thus revealing the membership of both samples. For text-to-image generation, I think one similar setting could be, find the similar images but with a significant text embedding difference, analyzing such specific correlated pairs could be helpful in understanding the impact. I think this is different from the training data deduplication.
> > > > > >
> > > > > > Adaptive attacks in Appendix A.5. Thank you for providing the adaptive attack design. Table 9 shows the trend that TPR increases as the number of generation rounds increase and if seems to me the max number of generations rounds could be $n$ while Table 9 presents the up to 3 rounds, which may not be sufficient.
> > > > > >
> > > > > > > Utility evaluation and intuitive explanation.
> > > > > >
> > > > > > Thank you for providing the additional results for CIFAR10 and Pokemon. This is helpful in understanding the utility impact of the design choice. Thanks for providing the explanation.
> > > > > >
> > > > > > > Writing such as Memorization and MIAs.
> > > > > >
> > > > > > Thank you for providing the clarifications.
> > > > > >
> > > > > > [R6] Dhariwal et al.  Diffusion Models Beat GANs on Image Synthesis. NeurIPS 2021.

---

> > > > > > > ### Author Response · Authors · 2024-12-03
> > > > > > > **Thank you**
> > > > > > >
> > > > > > > Dear Reviewer wt6x,
> > > > > > >
> > > > > > > Thank you for your time and efforts to review our paper. Your thoughtful feedback during the review and discussion helped us strengthen the paper significantly.
> > > > > > >
> > > > > > > Best regards,
> > > > > > >
> > > > > > > The Authors

---

### Official Review · Reviewer_bDxX · 2024-11-07

**Soundness:** 3
**Presentation:** 3
**Contribution:** 2
**Rating:** 6
**Confidence:** 4

**Summary:**

This paper presents a comprehensive study on a defense mechanism against membership inference attacks specifically targeting diffusion models. The authors employ an ensemble of two diffusion models, each trained on two disjoint datasets, to develop their defense strategy. Subsequently, they propose two distinct defenses utilizing these two models.
The first defense, referred to as DualMD, involves alternating between the two models during the diffusion process. By doing so, half of the steps are executed without any influence from half of the training data. This approach is expected to significantly reduce the overall impact of the training data, thereby making membership inference more challenging, particularly for black-box attacks.
The second defense, denoted as DistilMD, involves distilling the resulting model from DualMD into a new model. This process aims to eliminate even white-box attacks by further reducing the vulnerability of the model to membership inference attacks.
The authors provide empirical evidence demonstrating the effectiveness of their proposed defense mechanism in reducing the performance of existing white-box and black box attacks.

**Strengths:**

- Membership inference and memorization is a key challenge with diffusion models. Mitigating such vulnerabilities is hence an important topic.

- The idea of using an ensemble of model is clever.

**Weaknesses:**

- Although the idea is clever the same idea is explored in previous work as a way to defend against MIA both in classification [1] and generative models (diffusion models in particular!) [2]

- The paper is not evaluated against any adaptive attacks. Is there an attack that can be optimized for DualMD and DistillMD?

- The evaluation of utility is limited.

[1] https://www.usenix.org/system/files/sec22-tang.pdf

[2] https://openreview.net/forum?id=vuVGcl0ed1#all

**Questions:**

- The paper in [2] uses a very similar idea for realizing differentially private diffusion models.


- The utility of your models need to be evaluated more extensively. First, you need to provide some real generation so that reviewers can inspect the visual effect of your method. You also need more quantitive evaluation of your technique. It seems you are improving the metrics over non-private models? Does it mean you can claim SOTA for generating with diffusion models?

- I think there is a need for exploring adaptive attacks. For example, is there something the attacker can do by calling the decoding with different number of steps? If you believe the existing attacks are optimal against your defense, you need to bring a convincing argument.

---

> ### Author Response · Authors · 2024-11-21
> **Response to Reviewer bDxX (Part 1)**
>
> We thank the reviewer for the comments, we address main concerns as follows:
>
> 1. __Regarding the similarity with the papers [1] and [2]__: While these works look similar to ours in the way they divide datasets into subsets, they both have limitations and our paper presents unique insights of the vulnerability into diffusion models.
>
> - Firstly, while both papers employ an output aggregation framework to get the final results from the sub-models, this approach is not suitable for generative modeling tasks. Take the case when two models are trained on two non-identical subsets. If the two subsets are drawn from a same distribution but are not representative enough for that distribution, the two resulting models can follow two different trajectories in their generation process. Consequently, their outputs for the same input can be different, making the aggregation process become mixing two distinct images. There is no guarantee that the combination of the two images is a good image which should be evidenced by good generation scores. To validate this analysis, we tried to generate images by averaging the outputs of our two sub-models trained on two disjoint subsets of CIFAR10, and the resulting FID score in compared to our approach is reported in the table below.
>
> | Method          | FID  ($\downarrow$)  | IS  ($\uparrow$)  |
> | --------        | ---    | ---   |
> | Original Model  | 14.127 | 8.586 |
> | Aggregation     | 21.360 | 8.007 |
> | DistillMD (Our) | 14.192 | 8.391 |
>
> - Secondly, while the paper [2] is designed for diffusion models, the method is only tested with unconditional diffusion models on small datasets, including MNIST and CIFAR10. In contrast, we validated our method’s efficacy on both unconditional and text-guided diffusion models, shedding new insights into the difference between the two tasks. Specifically, our experiments suggest that DistillMD or DualMD alone cannot effectively defend MIAs for text-guided generation, which implies that similar methods in [1] and [2] are also not suitable for large text-to-image models because they all rely on dividing the datasets. Nevertheless, we investigated the issue and realized that the main cause is prompt overfitting, consequently proposed to diversify the training prompts of text-to-image diffusion models. Our investigation provided new insights into conditional diffusion models as we are the first ones to realize the difference between unconditional and conditional diffusion models. In addition, employing prompt diversification alongside DualMD and DistillMD further improves the defense capability of resulting models.
>
> - In addition, we provide new insights into the connections between MIAs and the model memorization issue. By proving that DistillMD not only defends MIAs but also mitigates memorization, we unify the two domains and simplify the defending mechanism of both to one common algorithm.
>
> - Besides, [1] is not explicitly designed for diffusion models, and there is no guarantee that their method will work well for all types of diffusion models. Besides, they propose to divide the datasets into multiple subsets and train multiple models on joint clusters of subsets. The large number of models can hinder the utility of the method for large-scale diffusion models because each model requires substantial memory to store, and train and running multiple models would be significantly slow. Hence, we propose reducing the algorithm to training only two models on two disjoint subsets, which lowers the memory consumption and preserving the property that for any training sample, there is always a model that "sees" it as a "test example".

---

> ### Author Response · Authors · 2024-11-21
> **Response to Reviewer bDxX (Part 2)**
>
> 2. __Regarding further utility evaluation__: We would like to provide some generated images by our model on CIFAR10 and Pokemon in the Appendix for reference. We also provide more quantitative results of the sub-models and more experiments with a new dataset Naruto [3] and a new architecture Stable Diffusion v2.1. For the reviewer's inquiry, our evaluation on image generation only gives comparable results with the baseline non-private models, so we are not claiming SOTA for image generation quality as it is not the main goal of our work.
>
> For Pokemon dataset with backbone SDv2.1:
> | Method         | FID ($\downarrow$) | IS ($\uparrow$) |
> |:-------------- | ------------------ | --------------- |
> | Original model | 0.44               | 2.99            |
> | DistillMD      | 0.41               | 3.55            |
>
>
> | Attack | Method     | AUC  | TPR@1%FPR ($\downarrow$) |
> |:------ | ---------- | ---- | --------- |
> | SecMIA | No defense | 0.98 | 0.189     |
> |   SecMIA     | DistillMD  | 0.54 | 0.019     |
> | PIA    | No defense | 0.45 | 0.012     |
> |   PIA     | DistillMD  | 0.47 | 0.007     |
>
> For Naruto dataset with backbone SDv1.5 and SDv2.1:
> | Backbone | Method         | FID ($\downarrow$) | IS ($\uparrow$) |
> | -------- |:-------------- | ------------------ | --------------- |
> | SDv1.5   | Original model | 0.18               | 2.16            |
> | SDv1.5   | DistillMD      | 0.22               | 2.19            |
> | SDv2.1   | Original model | 0.16               | 2.30            |
> | SDv2.1   | DistillMD      | 0.20               | 2.35            |
>
> | Backbone | Attack | Method     | AUC  | TPR@1%FPR ($\downarrow$) |
> | -------- |:------ | ---------- | ---- | --------- |
> | SDv1.5   | SecMIA | No defense | 0.93 | 0.475     |
> | SDv1.5   | SecMIA | DistillMD  | 0.46 | 0.005     |
> | SDv1.5   | PIA    | No defense | 0.45 | 0.007     |
> | SDv1.5   | PIA    | DistillMD  | 0.48 | 0.006     |
> | SDv2.1   | SecMIA | No defense | 0.90 | 0.333     |
> | SDv2.1   | SecMIA | DistillMD  | 0.45 | 0.006     |
> | SDv2.1   | PIA    | No defense | 0.47 | 0.008     |
> | SDv2.1   | PIA    | DistillMD  | 0.48 | 0.008     |
>
> 3. __Regarding Adaptive Attack concerns__: We thank the reviewer for the suggestion. We conduct an adaptive attack against DualMD by manipulating the number of steps as follows. We first call the model to generate an image in n steps; the input noise is denoised by the SubModel1 (SB1), then SubModel2 (SB2), and repeat. We add noise to the generated images to the second largest timestep to eliminate the effect of all denoising steps except the earliest one from SB1. The noisy image and its corresponding noisy timestep then serve as the initial noise and starting timestep for the second round of generation with n-1 steps starting again from SB1. The process can be repeated with more generation rounds to gradually remove the effect of SB2 and keep only denoising steps from SB1. Combining this generation process with black-box MIAs we have a complete attack, and the performance of DualMD against this attack is given in the table below. It can be seen that our method still maintains a high defense efficiency after rounds of this adaptive attack.
>
> | # rounds  | AUC    | TPR@1%FPR ($\downarrow$) |
> | --------  | ---    | ---       |
> | 2         | 0.53   | 0.024     |
> | 3         | 0.51   | 0.048     |
>
>
> We really hope that the reviewer can reconsider the review score. Please let us know if you would like us to do anything else.
>
> Best,
>
> Authors
>
> __References:__
>
> [1] Tang, et al. "Mitigating Membership Inference Attacks by Self-Distillation Through a Novel Ensemble Architecture". Proceedings of the 31st USENIX Security Symposium.
>
> [2] Sehwag, et al. "Differentially Private Generation of High Fidelity Samples From Diffusion Models".
>
> [3] Cervenka, Eole. "Naruto BLIP captions".

---

> > ### Author Response · Authors · 2024-11-29
> >
> > Dear Reviewer bDxX,
> >
> > Thank you for your thoughtful feedback on our paper. We have carefully addressed all the mentioned concerns, including the similarity with the papers [1] and [2], further utility evaluation (by adding new quantitative results of the sub-models and more experiments with a new dataset Naruto [3] and a new architecture Stable Diffusion v2.1), and additional experimental results and discussion on adaptive attack. We understand that you might be busy now, and we would deeply appreciate it if you could look at our rebuttal.
> >
> > Having further discussions helps achieve consensus and clarify misunderstandings, so we are eager to know if there are any remaining questions. We are more than happy to address any other follow-up discussions. Your further opinions are significant for evaluating our revised paper, and we look forward to hearing from you soon.
> >
> > Best regards,
> >
> > The Authors.

---

> > > ### Comment · Reviewer_bDxX · 2024-12-02
> > > **Thank you for the rebuttal**
> > >
> > > I thank the authors for their rebuttal. Most of my concerns have been resolved. Although I'm still not entirely excited about the novelty of this work, I believe the execution and experimentation are now strong, and I am raising my score.

---

> > > > ### Author Response · Authors · 2024-12-03
> > > > **Thank you so much for upgrading the paper score!**
> > > >
> > > > We thank Reviewer bDxX for the post-rebuttal response, especially for upgrading the paper score to 6. We greatly appreciate that! We are so glad that most of the concerns have been adequately addressed.
> > > >
> > > > Best,
> > > >
> > > > The Authors

---

### Author Response · Authors · 2024-11-25
**Regarding the changes in the rebuttal revision**

We sincerely thank the reviewers for their valuable feedback. In response, we have prepared a revision with the following key updates:

- A modification of Fig. 1's caption (Fig. 1).
- A transparent transition from MIAs to model memorization (Introduction).
- A more precise claim about the connection between MIAs and model memorization (Introduction).
- Additional experimental results on the Naruto dataset and Stable Diffusion 2.1 model (Table 2, Table 4 and Table 7).
- Qualitative examples generated by our methods on CIFAR10 and Pokemon (Fig. 5, Fig. 6 and Fig. 7).
- ROC curves for the black-box attack (Fig. 4).
- A proposed adaptive attack against DualMD and the corresponding analysis (Appendix A.5).
- A comparison with the ordinary knowledge distillation, which only employs one teacher model for the entire training dataset (Appendix A.6).

We believe these revisions effectively address the reviewers’ concerns and significantly improve the quality and clarity of the manuscript. Thank you for your insightful feedback.

---

### Meta-Review · Area_Chair_4ewA · 2024-12-22

**Metareview:**

This paper studies the membership inference defense for diffusion models by training two diffusion models on disjoint data. This work introduce two methods, DistillMD and DualMD. After training model A on dataset A and model B on dataset B, DistillMD uses model A to generate image for dataset B and model B to generate image for dataset A. DistillMD then use generated images to train a student model. DualMD use the model A and B to perform denoising together in denoising the images. At each step, DualMD use one model to perform denoising at that step, then at next step, DualMD use the other model to perform the denoising. Experimental evaluation include unconditional diffusion model and text-to-image diffusion model, including black-box attacks and white-box attacks.

This paper is on the borderline. Although their experiments got strengthened during the rebuttal period, I agree with the two reviewers in that the paper lacks methodological novelty.

**Additional Comments On Reviewer Discussion:**

The authors provided extra experimental results to rebut, and as a result, their experiments got strengthened during the rebuttal period.

---

### Decision · Program_Chairs · 2025-01-22

Reject